# Changes in the Release of Endothelial Extracellular Vesicles CD144+, CCR6+, and CXCR3+ in Individuals with Acute Myocardial Infarction

**DOI:** 10.3390/biomedicines12092119

**Published:** 2024-09-18

**Authors:** Alexa Moreno, Pedro Alarcón-Zapata, Enrique Guzmán-Gútierrez, Claudia Radojkovic, Héctor Contreras, Estefanía Nova-Lampeti, Felipe A. Zúñiga, Llerenty Rodriguez-Alvárez, Carlos Escudero, Paola Lagos, Claudio Aguayo

**Affiliations:** 1Department of Clinical Biochemistry and Immunology, Faculty of Pharmacy, University of Concepcion, P.O. Box 237, Concepción 4030000, Chile; alexa.moreno@unesum.edu.ec (A.M.); pedroalarcon@udec.cl (P.A.-Z.); eguzman@udec.cl (E.G.-G.); cradojkovic@udec.cl (C.R.); heccontr@udec.cl (H.C.); enova@udec.cl (E.N.-L.); fzuniga@udec.cl (F.A.Z.); paolalagos@udec.cl (P.L.); 2Clinical Laboratory Program, Faculty of Health Sciences, State University of Southern Manabí, Jipijapa 130402, Ecuador; 3Department of Animal Science, Faculty of Veterinary Sciences, University of Concepcion, Chillán 3780000, Chile; llrodriguez@udec.cl; 4Vascular Physiology Laboratory, Department of Basic Sciences, Universidad del Bio-Bio, Chillán 3780000, Chile; cescudero@ubiobio.cl; 5Group of Research and Innovation in Vascular Health (GRIVAS Health), Chillán, 3780000, Chile

**Keywords:** myocardial infarction, extracellular vesicles, CD144

## Abstract

Acute myocardial infarction (AMI) results from vulnerable plaque rupture, causing ischemic cardiomyocyte necrosis and intense inflammation. Paradoxically, this inflammation releases factors that aid heart repair. Recent findings suggest a role for extracellular vesicles (EVs) in intercellular communication during post-AMI cardiac repair. However, EVs’ tissue origin and chemokine profile in the blood of patients with AMI remains unclear. This study characterized the tissue origin and chemokine receptor profile of EVs in the coronary and peripheral blood of patients with AMI. The results reveal that vesicles isolated from coronary and peripheral blood plasma are enriched in tetraspanin (CD9) and express CD81^+^, CD90^+^, and CD144^+^. The vesicle size ranged between 145 and 162 nm, with the control group exhibiting smaller vesicles (D10) than the AMI group. Furthermore, all vesicles expressed CCR6 and CXCR3, whereas a small percentage expressed CCR4. In addition, a decrease in CXCR3 and CCR6 expression was observed in coronary and peripheral AMI blood vesicles compared with the control; however, no difference was found between AMI coronary and AMI peripheral blood vesicles. In conclusion, our study demonstrates, for the first time, changes in the number of extracellular vesicles expressing CD144^+^, CXCR3, and CCR6 in the peripheral circulation of patients with AMI. Extracellular vesicles present in the circulation of patients with AMI hold excellent promise as a potential diagnostic tool.

## 1. Introduction

Acute myocardial infarction (AMI) arises from complete or partial artery obstruction instigated by the rupture or erosion of a vulnerable plaque [1]. This event leads to cardiomyocyte death [2], a critical consequence considering the limited regenerative capacity of the adult mammalian heart. Therefore, AMI is characterized by losing a significant amount of myocardium, replaced by a collagen-based scar, ischemic cardiomyocytes, and a prominent inflammatory reaction. These elements are pivotal in heart repair and remodeling [3].

Extracellular vesicles (EVs) are classified into three subtypes based on size and biogenesis: exosomes, microvesicles, and apoptotic bodies. Among them, exosomes are the smallest (~100 nm in diameter), derived from the endosomal pathway, and they carry nucleic acids, proteins, lipids, and metabolites. This cargo facilitates near- and long-distance intercellular communication regarding health and disease [4,5,6]. Evidence indicates that EVs can attenuate cellular senescence, inflammation, and myocardial injury [7,8], offering hope for their potential therapeutic applications. With the emergence of studies on exosome secretion by various cell types, including cardiomyocytes (CMs), endothelial cells (ECs), fibroblasts, and circulating progenitor cells (CPCs), protective effects on AMI patients receiving EVs have been demonstrated [9].

Recent findings also suggest the involvement of exosome-mediated intercellular communication in cardiac repair after AMI [10]. It has been shown that EVs are secreted by diverse cardiac stem cells and are associated with cardioprotective effects, including activating regenerative signals (angiogenic and anti-inflammatory) and active participation in cardiac repair [11]. For instance, Chengwei et al. [12] revealed that EV release from cardiac mesenchymal stem cells promotes repair in AMI. Moreover, they found that the transfer of EV molecular cargo, including miRNA, contributes to cardioprotection in a mouse AMI model. EVs isolated from these cardiac mesenchymal stem cells were injected intramyocardially into the left ventricular wall (border zone) at three locations immediately after the left anterior descending ligation. Additionally, EVs isolated from mesenchymal cells promoted neovascularization and reduced infarct size after intramyocardial administration in a porcine model of chronic myocardial ischemic damage [13]. EVs released from mesenchymal stem cells can directly reach the ischemic tissue, participating in cardiac regeneration and remodeling [14,15]

Given their ease of retrieval from biological fluids like serum, plasma, and urine, EVs have also emerged as potential biomarkers for the evolution of AMI [7,8]. However, the tissue origin and chemokine receptor profile of extracellular vesicles in the coronary and peripheral blood of patients with AMI remain unknown [16,17]. Understanding their origin is critical, since several cardiovascular risk factors have been associated with the differential cargo of circulating EVs [18], which differentially activate target cells [19] via the transference of their cargo (proteins, mRNAs, miRNAs) to recipient cells. Therefore, the chemokine content in circulating EVs is critical to understanding its potential beneficial effect in patients with AMI.

This study aimed to characterize the tissue origin and chemokine receptor profile of plasma-derived EVs in the blood of patients with AMI.

## 2. Materials and Methods

### 2.1. Subjects

Patients treated for AMI at the Centro Cardiovascular del Hospital Clínico Dr. Guillermo Grant Benavente were included based on specific diagnostic criteria. AMI patients presented ST-segment elevation AMI within less than 24 h of onset, regardless of whether they underwent primary percutaneous coronary intervention or thrombolysis [20]; chest pain lasting more than 20 min with electrocardiographic confirmation; elevated levels of troponin T and I; and creatine kinase-myocardial band (CKMB) isoenzyme. Additional inclusion criteria for participants include the following: (i) adults up to 80 years of age, of either sex, (ii) with or without prior treatment using acetylsalicylic acid, beta-blockers, angiotensin-converting enzyme inhibitors, or statins. Exclusion criteria: (i) Individuals with a history of previous myocardial infarction. (ii) Renal and hepatic impairment. (iii) Coagulation disorders. (iv) Moderate to severe heart failure from another cause, not ischemic. (v) Active infectious or inflammatory processes. (vi) Chronic inflammatory disease. (vii) Malignant disease. (viii) Recent trauma or surgical interventions. (ix) Blood transfusions in the last ten months.

As control subjects, we recruited age-matched individuals from the Universidad de Concepción who met the following inclusion criteria: (i) Healthy subjects; (ii) adults (up to 80 years) of both sexes; and (iii) without a history of minor or significant cardiovascular events. Also, we used the following exclusion criteria: (i) individuals with hypotension and infectious or inflammatory symptoms and those (ii) consuming anticoagulants or any non-steroidal anti-inflammatory drug. (iii) There was a blood donor during the week preceding the sampling. (iv) Consuming excess vitamin supplements resulted in exclusion, as did (v) pregnancy.

This study was approved by the Ethical Committee of the Servicio de Salud Concepción, Chile, and all included subjects signed the corresponding informed consent form (Approval Code: 17-08-47, 16 January 2018).

### 2.2. Echocardiography

The echocardiographic analysis was performed with a Vivid seven ultrasound machine (General Electric, Healthcare, Horten, Norway) with the patient in the left lateral position within 48 h after AMI. The left ventricular volume and ejection fraction was calculated using images from an apical 4-chamber view. Two cameras were analyzed using the biplane Simpson’s method [21]. The left ventricle (LV) was evaluated along the division the American Society of Echocardiography recommendations, investigating the regional systolic function [22].

### 2.3. Samples and Analytical Procedures

Venous blood was drawn from the antecubital vein. Blood samples were collected in tubes containing EDTA. The samples were centrifuged at 1500× *g* for 15 min at 4 °C, the plasma at 4 °C, and used within 24 h for glucose and lipid profiles, and serum aliquots at −70 °C.

Glucose, triglycerides, and total cholesterol were quantified by standardized methods (Roche Diagnostics, Mannheim, Germany) in a Hitachi 917 autoanalyzer (Tokyo, Japan). The low-density lipoprotein-cholesterol (LDL-C) level was determined as the difference between total cholesterol and the cholesterol contained in the supernatant obtained after the selective precipitation of LDL with polyvinyl sulfate in polyethyleneglycol. HDL was isolated in the supernatant obtained following the precipitation of apolipoprotein (apo) B-containing lipoproteins with phosphotungstic acid in the presence of magnesium ions. Quality control was performed using RIQA Program (Ireland). In the isolated LDL fractions, triglycerides and cholesterol were measured using the previously mentioned methods, phospholipids using the Bartlett method, and proteins using the Lowry method [23]. According to the manufacturer’s instructions, duplicate measurements of the LOX-1 levels were performed using an ELISA kit (Life Science Inc., USCN, Wuhan, China). The minimum detectable level of human LOX-1 is less than 0.04 ng/mL. C-reactive protein (CRP) levels were determined using an ILAB 600 autoanalyzer (Roche Diagnostics, Mannheim, Germany) using a Quantex CRP Ultra-Sensitive commercial kit (Biokit, S.A., Barcelona, Spain), following the manufacturer’s instructions.

### 2.4. Isolation and Characterization of Human Plasma EVs

Venous blood samples were collected in tubes containing EDTA, aliquoted in 700 μL volumes, and stored at −20 °C until use. Frozen plasma samples were thawed and centrifuged at (i) 2000× *g* for 30 min at 4 °C and (ii) 12,000× *g* for 45 min at 4 °C, and then passed through a 0.22 μm CA syringe filter to remove cell debris and large extracellular vesicles, and then used for the isolation of small EVs by size exclusion chromatography, as described in Contreras et al., 2023 [24].

### 2.5. Size Exclusion Chromatography (SEC)

For size exclusion chromatography (SEC), 500 μL of concentrated plasma was loaded onto the Sephadex column (G200/120 or G200/40 resin, Sigma Chemical Company, Saint Louis, MO, USA). Once the sample reached the top of the column bed, elution with NaCl solution was initiated, and the eluate was collected by gravity. Each fraction was collected in volumes of 400 μL for the EV collection medium. The 14 fractions 14 were collected and stored at −20 °C until use. The total protein concentration was determined for each fraction using the DC TM Protein Assay Kit (BioRad Laboratories, Hertfordshire, UK), following the manufacturer’s instructions.

After each separation, the SEC columns were washed sequentially with 30 mL of 0.1 M NaOH solution (0.22 μm filtered and degassed by sonication), 60 mL of PBS, and 100 mL of filtered and degassed mobile phase for reuse. The fractions collected from the columns were subjected to gel electrophoresis with Coomassie Blue Staining. The samples were loaded onto 6% tris-glycine-SDS gels and electrophoresed at 180 V and 40 mA for 100 min. Subsequently, the gels were stained with the Coomassie Stain at room temperature for 2 h, followed by the removal of excess stain by washing with a bleaching solution (20% methanol and 10% acetic acid in distilled water).

### 2.6. Extracellular Vesicles Characterization

For Western blot analysis, 40 μg of non-standardized protein homogenized from fraction 1 to 8 (1–8) was separated using 10% SDS/PAGE gel and transferred to PVDF membranes. The membranes were incubated overnight at 4 °C with mouse anti-ALIX and mouse anti-CD9 purchased from Santa Cruz Biotechnology. Then, the membranes were washed and incubated with secondary antibodies, as previously described [24].

The size distribution and concentration of isolated vesicles were measured using a NanoSight NS 300 instrument (Malvern Instruments Ltd., Malvern, UK), and the data were analyzed using NTA software (version 3.2 Dev Build 3.2.16) as previously described [24].

### 2.7. Flow Cytometry of EVs

To evaluate the enrichment of EVs isolated from the column, a flow cytometry analysis of the fractions was performed using the Exosome-Human CD63 Isolation/Detection Kit (Invitrogen, Carlsbad, CA, USA), according to the manufacturer’s recommendations, and as previously described [24]. Briefly, 25 μg EVs was incubated with 20 μL beads (Life Technologies, Carlsbad, CA, USA) overnight at 4 °C. Next, glycine diluted in PBS (100 mM final concentration) was added, mixed gently, and kept for 45 min at room temperature. The EV/bead complexes were washed twice with 1 mL of PBS/0.5% bovine serum albumin by centrifugation at 1500× *g* for 3 min at RT. The EV/bead complexes were incubated with primary FITC-conjugated CD81 (Abcam,, Cambridge, UK; catalog no. 34162, clone MM2/57), CD90 (Abcam, Cambridge, UK; catalog no. 95700, clone 5E10), CD144 (Abcam, Cambridge, UK; catalog no. 272346, clone 55-7h1), CCR4 (Abcam, Cambridge, UK; catalog no. 281322, clone EPR235002-85), CCR6 (Abcam, Cambridge, UK; catalog no. 288307, clone EPR235002-85), or CXCR3 (Abcam, Cambridge, UK; catalog no. 314293, clone EPR23845-44) for 1 h at room temperature. A negative control antibody reaction was performed using latex beads, incubated with anti-CD63 or anti-CD81 for 1 h at room temperature (RT). The labeled MV/bead complexes were pelleted and washed twice as above with 300 μL of PBS/0.5% bovine serum albumin. Finally, 100 μL pellets were resuspended in 200 μL of focusing fluid and subjected to flow cytometry using BD LSR Fortessa™ X-20 (BD Biosciences, NJ, USA).

### 2.8. Statistical Analysis

The data were analyzed using standard statistical software (SPSS version 25) and GraphPad Prism 9.0 (GraphPad Software Inc., San Diego, CA, USA). The values are expressed as means and S.E.M. The variables were analyzed using non-parametric ANOVA tests. Mann–Whitney tests were used for pair comparisons in cases with significant differences (*p* < 0.05).

## 3. Results

### 3.1. Studied Group

There were no differences in the average age, weight, height, waist circumference, and cholesterol levels between the controls and the AMI patients. However, statistically significant differences were found for BMI, glycemia levels, and sLOX-1 (*p* < 0.05 in all comparisons). Also, following the diagnosis criteria, patients with AMI exhibited elevated cardiac markers (Troponin, CK-Total, and CK-Mb) compared to the control group (Table 1). In contrast, lipid values, C-reactive protein (CRP), creatinine, and prothrombin time remained within normal ranges.

Extracellular vesicles were isolated from the plasma of healthy individuals (HC-P) by centrifugation and size exclusion chromatography. Eluted fractions were collected, and the total protein concentration was measured using a BSA standard curve (Figure 1A). Extrapolating the concentration values of the obtained fractions reveals that the total protein concentration peaks at 10 mg/mL between fractions 4, 5, and 6, with fraction 6 exhibiting the highest protein enrichment (Figure 1A). Figure 1B shows a gel-electrophoresis stained with Coomassie Blue, showing that fractions 4, 5, and 6 contain the highest concentrations of proteins. Due to their enriched protein content, fractions 1–8 were specifically selected for further analysis with Western blotting.

Subsequently, the content was transferred to a membrane, and Western blotting was performed for Alix, Tsg101, and CD9. As shown in Figure 1C, fractions 3–6 display signals for Alix and CD9. Fractions 4, 5, and 6 were enriched with EVs expressing all three proteins. At the same time, no expression of ALIX, Tsg101, or CD9 proteins was detected in the remaining fractions.

For the characterization, we used the Exosome-Human CD63 Isolation/Detection kit [23] and labeled the fractions with anti-CD81, anti-CD90, and anti-CD144 (Figure 1D). Non-significant differences were noted in this signal across markers between the analyzed fractions (fractions 4, 5, and 6), with the signal being most abundant in fraction 5 in the flow cytometry analysis of these fractions (Figure 1D). Therefore, we used fractions 4 to 6 for the rest of the experiments.

### 3.2. Characteristics of Peripheric and Coronary EVs from AMI (MI-P and MI-C) and Controls (HC-P)

We then counted the number of the isolated EVs and measured their size using NTA in the fractions pool (F4, F5, F6). EVs from the control group (HC-P) have 3.2 × 10^11^ particles/mL with a modal size of 106.3 ± 3 nm (*n* = 5). In contrast, in the coronary blood samples (MI-C) from patients with AMI, the concentration of EVs was significantly higher, 9.1 × 10^11^ particles/mL, without changes in the modal size of 112.8 ± 4 nm (*n* = 8). In addition, EVs from the peripheral blood group of patients with AMI yielded 1.0 × 10^12^ particles/mL with a modal size of 121.4 ± 5 nm (Figure 2B) (*n* = 5). We then separated the sizes of the particles in the 10th, 50th, and 90th percentiles. Interestingly, for the 10th percentile, there were significant differences in the size of the EVs when comparing those isolated from the coronary blood of patients with AMI (MI-C) versus that of those in the control group (HC-P) (Figure 2C). The control group’s values were significantly smaller (92 ± 1.6 nm) than those isolated from the coronary blood of AMI patients (MI-C, 100 ± 2 nm), and there were no significant changes compared with those isolated from the peripheral blood of AMI patients (MI-P). In addition, at the 50th percentiles, EVs isolated from the control group were significantly smaller (124 ± 3 nm) than those isolated from peripheral blood (141 ± 3 nm), without significant changes compared with those isolated from the coronary blood of AMI patients (MI-P, 134 ± 1 nm) (Figure 2C). No significant differences were found in the 90th percentile of EV size between control and AMI samples (coronary or periphery blood samples) (Figure 2C).

### 3.3. Immunophenotype Characterization of Extracellular Vesicles by Flow Cytometry

The EVs were isolated by size exclusion chromatography, incubated with the exosome-human CD63 isolation/detection kit, and labeled with anti−CD90, anti−CD81, and anti-CD144 antibodies. The EVs isolated from the plasma of control and AMI patients (coronary and peripheral blood) were positive for CD81, CD90, and CD144 labeling (Figure 3).

The individual analysis of these markers in CD63^+^ vesicles showed that CD81^+^ was differentially expressed in the EV from controls (17%) versus AMI patients, both in the coronary (24%) and peripheral (21%) source of the EVs (Figure 3), reaching statistical significance between the controls and the peripheral source of EVs. At the same time, no significant differences were found in CD81^+^ EVs from the coronary and peripheral sources in AMI patients.

Similar findings were found in the analysis of CD144^+^ (Figure 3). Thus, EVs from control samples have a lower proportion of CD144^+^ EVs (6%) compared with AMI patients, again reaching significant differences with the peripheral EVs of AMI patients (8%, *p* < 0.05).

Furthermore, most EVs isolated from controls and AMI patients (coronary or peripheral source) were CD90+ (Figure 3), showing non-significant differences in the ANOVA comparative analysis.

### 3.4. Characterization of the Chemokine Receptor Profile of Extracellular Vesicles by Flow Cytometry

The EVs were labeled with anti-CCR4, anti-CCR6, and anti-CXCR3 antibodies, as shown in the representative dot plots (Figure 4). CCR6 EVs represented 5 to 10% of those isolated from control and AMI patients. At the same time, only 0.3% to 0.5% were CCR4+ EVs in the studied groups. The proportion of CXCR3 EVs was significantly lower in the coronary and peripheral EVs from AMI patients than their counterparts in the control group. No significant differences were found in AMI patients’ EVs from the coronary and peripheral sources. The percentage of CCR6 EVs was significantly lower in the peripheral EVs from AMI patients than their counterparts in the control group. At the same time, no significant differences were found in CCR6 EVs from the coronary sources in AMI patients vs. the control group.

## 4. Discussion

Several studies have revealed that EVs released during AMI exhibit specific molecular signatures reflective of their tissue origin and pathophysiological state. Also, hypoxia, inflammation, or injury leads to the increased secretion of EVs by cardiomyocytes and other cardiac cells [6,7]. Our study demonstrates that AMI transiently increases circulating levels of CD144^+^ and decreases CCR6 and CXCR3 EVs. Indeed, those EVs have differential characteristics in their size, number, and proportion of CD81^+^ and CD144^+^ EVs compared to EVs isolated from control subjects. At the same time, MI-C EVs were not significantly different from MI-P, except in the expression of CCR6, in which the former have levels similar to those of the controls, while a significant reduction was found in the latter. Therefore, our results suggest a differential population of EVs in the periphery and cardiac origin after AMI. Thus, EVs from patients with AMI show high circulating levels, associated with differential size profiles depending on whether samples are collected from cardiac or peripheral circulation.

Our results align well with these previous studies that demonstrate that EV levels are elevated in patients with acute coronary syndrome (ACS), unstable angina (UA), and AMI when compared to stable coronary artery disease (SCAD), stable angina (SA), noncoronary controls, or healthy controls [25]. Also, high levels of endothelial cell EVs have been found in patients with coronary artery disease (CAD) [26], acute coronary syndrome [27], and myocardial infarction [28], representing a mechanism responsible for the dissemination of the procoagulant and proinflammatory signals to sites remote from the microenvironment of their formation.

Plasma-isolated vesicles were positive for CD90^+^, CD81^+^, and CD144^+^. Differences were observed only in the CD144^+^ samples isolated from the control and MI groups. These findings are consistent with the literature, indicating that under conditions of damage, such as AMI, there is an increased release of CD144^+^, a marker for endothelial dysfunction, since it is associated with VE-cadherin, a constitutive marker of endothelial cells [17,18]. Levels of CD144^+^ and CD31^+^/CD41^+^ endothelial microparticles (EMPs) serve as an index of endothelial injury in various diseases [29,30]. Thus, elevated levels of CD144^+^ EMPs may reflect endothelial dysfunction and structural vascular damage in ischemic cardiovascular diseases (ICVDs). Also, CD144^+^ EMPs are significantly associated with indices of neurological damage [28,29,30], supporting their utility as independent biomarkers of ICVD severity to assess the extent of ischemia. EVs from the endothelium are notable for their capacity to mirror endothelial dysfunction [29,30], correlate with cardiovascular outcomes, and predict disease severity [31] and plaque rupture in acute coronary syndrome patients [31]. Associations between circulating EVs and angiographic lesions have also been reported and correlated with a higher calculated 10-year Framingham Coronary Artery Disease risk [31,32,33,34]. Therefore, CD144^+^ EVs could be a marker of future ventricular arrhythmia and AMI, a biomarker of ischemic severity in the clinic, or predict cardiovascular events after AMI.

On the other hand, our work shows that in AMI, circulating EVs express chemokine receptors (CCR6 and CXCR3) and can be detected in the peripheral circulation of patients. Compared with controls, significant decreases were observed in AMI patients’ CCR6+ and CXCR3+ EVs percentages. Currently, there are no reports detailing the expression of any of these receptors on the surface of vesicles from patients with AMI. However, studies by Yang et al. (2019) show that CD4^+^/CXCR5/CXCR3 exosomes increase during acute rejection following renal transplantation, and suggest that EVs could promote the proliferation and differentiation of B cells, which play an essential role in developing antibody-mediated injections for renal transplantation [35].

CXCR3 is a G-protein coupled cell surface receptor that allows for functional selectivity on the tissue, receptor, and ligand level [6]. It binds three inflammatory chemokines: CXCL9, CXCL10, and CXCL11 [14,15]. CXCR3 ligands have also been studied in patients with left ventricular dysfunction and heart failure [36,37,38]. Circulating CXCL9, CXCL10, and CXCL11 levels were increased in subclinical and symptomatic left ventricular dysfunction, reaching statistical significance only in symptomatic patients [36]. In addition, CXCR3 ligands and established risk factors significantly improved the risk prediction models for left ventricular dysfunction [37]. Sanhueza et al. showed that CXCL9 and monocytes were augmented considerably in coronary blood during coronary infarction compared with peripheral blood, indicating that CXCL9 is an inflammatory mediator of vascular damage [39]. Therefore, CXCR3 and its ligand CXCL9 may constitute a biomarker for coronary infarction.

Extracellular vesicles (i.e., exosomes), as carriers of various signaling molecules, including chemokines like CXCR3, play a critical role in mediating intercellular communication during inflammation. Once activated in the inflammation site, leukocytes can uptake those EVs. This process may effectively reduce circulating levels of CXCR3^+^ exosomes in the peripheral blood.

On the other hand, the CCR6 receptor facilitates the release of inflammatory monocytes from the bone marrow into the bloodstream, mediated by the chemokine ligand CCL20 [40]. During atherogenesis, endothelial dysfunction elevates local CCL20 levels, directing CCR6 cells (B cells, immature dendritic cells, innate lymphoid cells (ILCs), regulatory CD4 T cells, and Th17 cells) to the subendothelial space [41,42]. The observed decrease in the percentage of CCR6+ EVs in peripheral blood from AMI patients compared to controls can be linked to the role of CCR6 in mediating inflammatory responses and tissue repair mechanisms following myocardial injury. CCR6 and chemokine CCL20 guide immune cells like T cells and lymphocytes to sites of inflammation, which is crucial in the context of AMI [43,44]. These immune cells play a significant role in the ischemia–reperfusion injury commonly observed in AMI. Normally, CCR6+ EVs might be expected to increase inflammatory conditions to facilitate the recruitment and activation of immune cells at the injury site. The observed decrease in the percentage of CCR6+ EVs in the peripheral blood of patients with AMI could be associated with the retention of these vesicles in the infarcted tissue. This hypothesis suggests that EVs carrying CCR6 might be sequestered and accumulated in the affected cardiac tissue during an AMI event, where they actively mediate inflammatory responses and tissue repair processes, reducing circulating EV levels.

Cardiac-specific markers and inflammatory molecules could provide valuable information for risk stratification and disease progression monitoring [30,33,35]. In addition to their diagnostic potential, EVs may play a role in the pathophysiology of AMI and cardiac repair processes. In contrast, EVs derived from cardiac or regenerative cells may carry factors that stimulate angiogenesis, tissue repair, and cardioprotection [45].

The size of our patient cohort limits our study. Additionally, it may not adequately capture the temporal variations in EV concentration and composition following a myocardial infarction. Additionally, we need to look further into the analysis of EV cargo. Furthermore, we also need to study the biological effect of EVs during a heart attack, and how they can contribute to cardiac repair.

In conclusion, our study demonstrates, for the first time, changes in the number of vesicles expressing CD144+, CXCR3+, and CCR6+ in the peripheral circulation of patients with AMI, suggesting that the presence of EVs in the peripheral circulation of patients with AMI holds excellent promise as a potential diagnostic tool. However, further research is needed to determine the pathophysiological role of these EVs in the development of AMI. Understanding how these vesicles contribute to the progression and severity of this disease could enhance their diagnostic and therapeutic potential, providing deeper insights into their involvement in myocardial infarction.

## Figures and Tables

**Figure 1 biomedicines-12-02119-f001:**
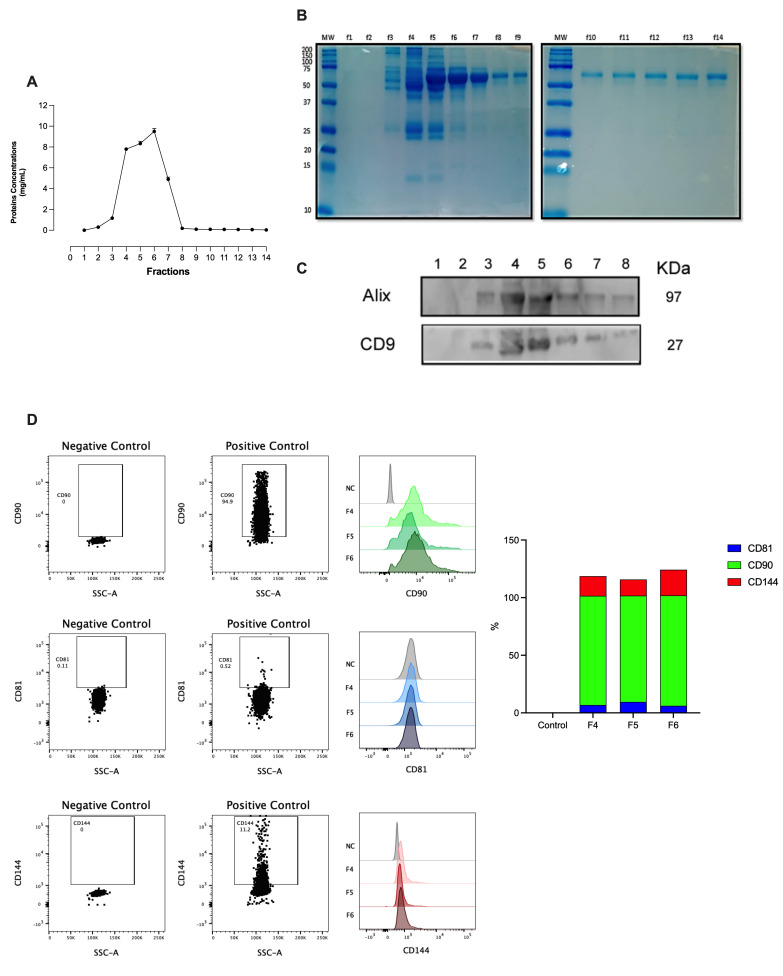
Protein profile fractions, Western blot, and flow cytometry analysis of EVs isolated from plasma fractions. Consecutive fractions were collected using size exclusion chromatography (SEC), as detailed in the methods section, to complete a 400 μL elution volume per fraction. Each combined fraction was concentrated in a centrifugal rotary evaporator at 4 °C overnight and resuspended in 200 μL. (**A**,**B**) Protein elution profile for the EV-collection medium. (**C**) Western blot analysis for the Alix, Tgs 101, and CD9 expression of the collected fractions from the column. (**D**) Small CD63-positive EVs were captured from fractions 4, 5, and 6, employing Dynabeads coated with anti−CD63 and stained with anti−CD90, anti−CD81, or anti−CD144. Negative controls were used for each antibody to ensure the specificity of the staining process. The left panel displays a representative image standardizing the identified vesicles’ positivity percentage. The right panel represents small EV bead CD63-positive staining for each antibody across the different fractions.

**Figure 2 biomedicines-12-02119-f002:**
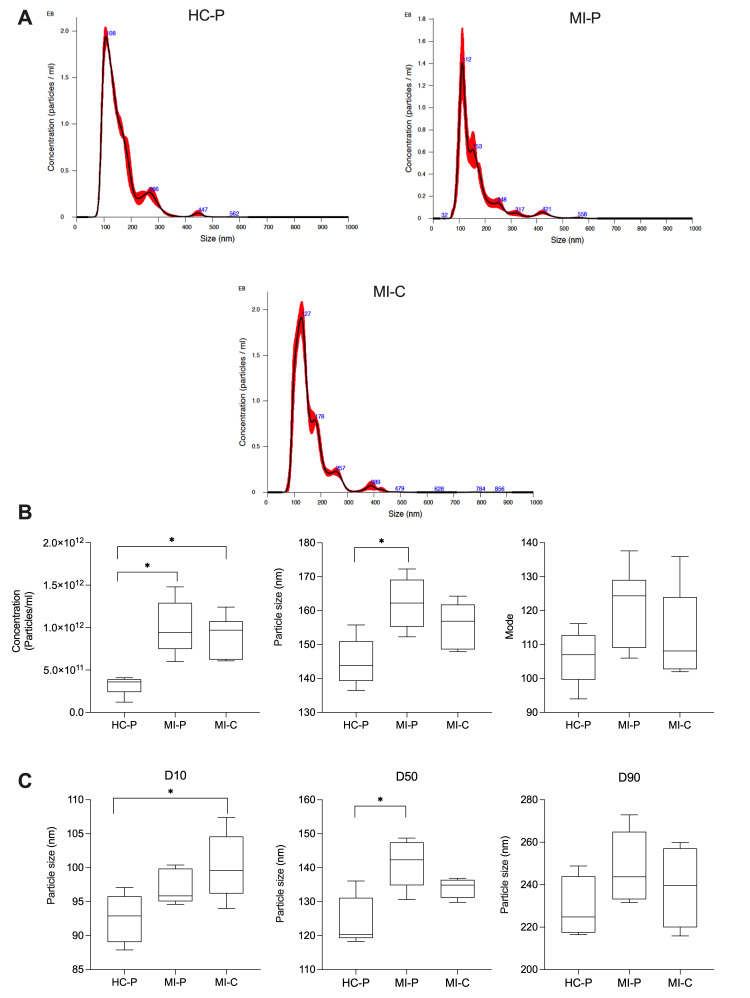
An analysis of the concentration and size distribution of EVs isolated from plasma. The NTA data show the size distribution of EVs isolated from the control group (HC–P), coronary blood (MI–C), and peripheral blood (MI–P) of patients with myocardial infarction (**A**). Bar graphs compare the concentration, mean size, and mode size of MVs in plasma (**B**). In (**C**), the size distribution of MV from plasma is shown as values for D10, D50, and D90. The values are means ± standard deviations (*n* = 5–8 in each group). * *p <* 0.05 versus HC-P.

**Figure 3 biomedicines-12-02119-f003:**
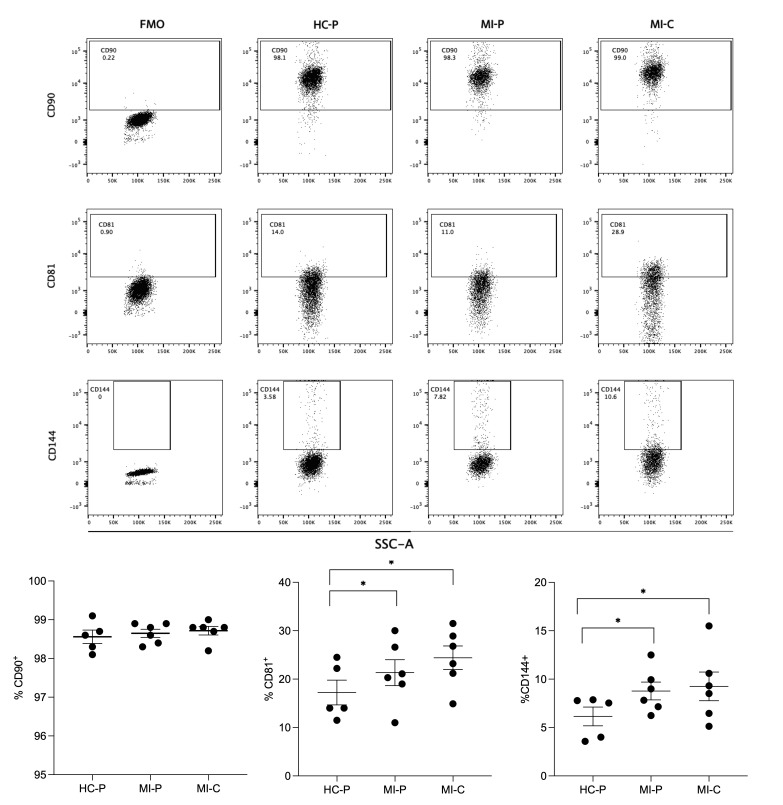
Flow cytometry characterizes derived EVs from fractions of plasma. In the upper panel, flow cytometry was used to identify CD90^+^, CD81^+^, and CD144^+^ antigens in EVs isolated from plasma from the control group (HC–P) and the coronary blood (MI–C) and peripheral blood (MI–P) of a patient with myocardial infarction (AMI). The lower panel bar graphs compare the % of CD90^+^, CD81^+^, and CD144^+^. The percentages are means ± standard deviations (*n* = 5–8 in each group). ** p <* 0.05 versus HC-P.

**Figure 4 biomedicines-12-02119-f004:**
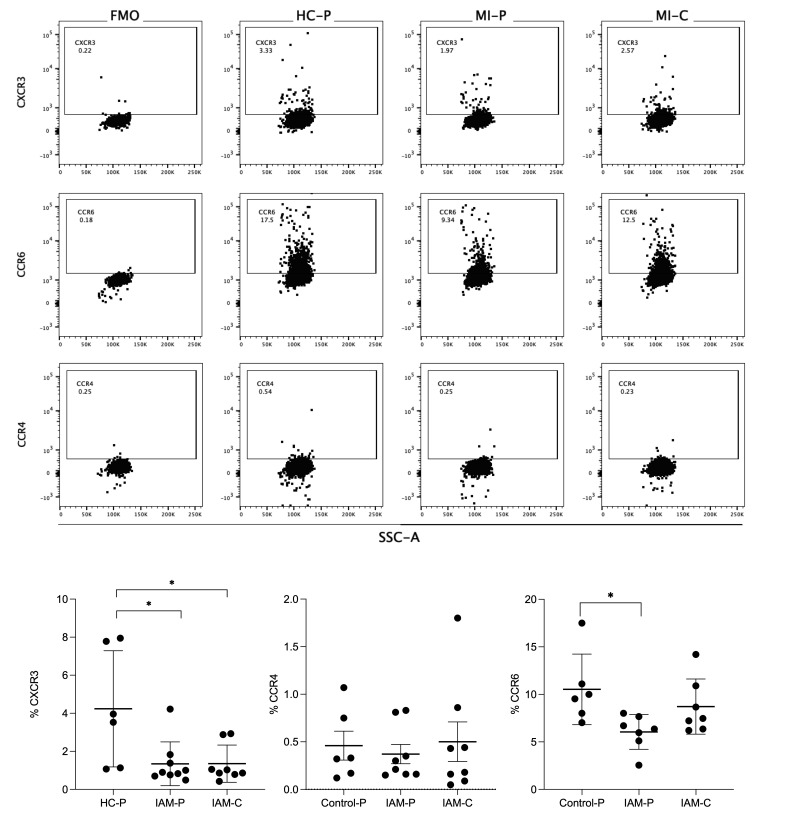
Flow cytometry of chemokine receptors express EVs isolated from fraction plasma. In the up panel, flow cytometry was used to identify CCR4, CCR6, or CXCR3 antigens in fractions of plasma from the control group (HC–P), coronary blood (MI–C), and peripheral blood (MI–P) of patients with myocardial infarction. The lower panel bar graphs compare the % CCR4, CCR6, or CXCR3. The percentages are means ± standard deviations (*n* = 5–8 in each group). ** p* < 0.05 versus HC-P.

**Table 1 biomedicines-12-02119-t001:** Clinical and biochemical characteristics from control subjects and acute myocardial infarction (AMI) patients.

	CONTROL	AMI
*n*	8	10
Gender (F/M)	5/3 (62%/38%)	6/4 (60%/40%)
Age	60 ± 2.4	64.8 ± 5.1
Weight (kg)	74.9 ± 5.7	70.1 ± 4.2
Height (m)	1.6 ± 0.1	1.6 ± 1.4
BMI	29.6 ±1.9	26.6 ±1.2 *
Waist Circumference (cm)	95.5 ± 4.6	102.6 ± 3.3
Glycemia (mg/dL)	76.4 ± 5.6	176.0 ± 45.0 *
Total Cholesterol (mg/dL)	171.7± 14.5	181.5± 12.7
Triglycerides	130.4 ± 7.6	149.6 ± 27.9
LDL Cholesterol (mg/dL)	119.3 ± 20.8	107.8 ± 12.4
HDL Cholesterol (mg/dL)	38.1 ± 5.1	43.0 ± 3.9
sLox-1	156.7 ± 55	294 ± 70.3 *
Troponin (ng/L)	12 ± 3.5	238 ± 72.4 *
CK-Total (U/L)	150 ± 20	2.205 ± 677 *
CK-Mb (ng/mL)	3.0 ± 3	219.8 ± 64.7 *
hCRP (mg/dL)	1.0 ± 0.2	1.6 ± 0.5
Previous Myocardial Infarction		
Yes	-	0 (0%)
No	-	10 (100%)
Diagnosis		
AMI with ST-segment elevation in inferior wall	-	5 (50%)
AMI with ST-segment elevation in anterior wall	-	4 (40%)
AMI with ST-segment elevation in anteroseptal area	-	1 (10%)
TIMI Score		
I	-	0
II	-	0
III	-	10 (100%)
KILLIP score		
I	-	5 (50%)
II	-	5 (50%)
III	-	0 (0%)
IV	-	0 (0%)

* *p* < 0.05 versus control.

## Data Availability

The original contributions presented in the study are included in the article, further inquiries can be directed to the corresponding author/s.

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
