# Peer review of "Changes in the Release of Endothelial Extracellular Vesicles CD144+, CCR6+, and CXCR3+ in Individuals with Acute Myocardial Infarction"

_biomedicines, 2024, doi:10.3390/biomedicines12092119_

Round 1

Reviewer 1 Report

Comments and Suggestions for Authors

In this paper authors report a work based on a deep characterization of extracellular vesicles (EVs) involved in Acute Myocardial Infarction (AMI), with a particular focus on tissue origin and chemokine profile of EVs in the blood of patients with AMI. The work is well constructed and a proper experimental design has been put in place to characterize expression of different markers on the EVs. Anyway, a major concern raises looking at part of the obtained results and the consequent discussion based on them. This is mainly related to differences founded in markers expression in control vs AMI patients and it is detailed in the major point reported here below. Moreover a complete revision of the text is required because there are many mistakes (reported in the “Minor points”) that somehow denote a little bit of superficiality in the paper preparation.

Major points:

1-    For some of the provided results there is no agreement between what is reported in the graphs and in the discussion. In particular: 1- results reported in Figure 4 clearly say that CXCR3 and CCR6 expression is lower in both MI-P and MI-C groups than in HC-P (control group). Despite this evidence, both in the result and in the discussion sections authors reported that these two markers are lower in the control group. This is important to be clarified because large part of the discussion is based on this comparison which is clearly contradictory. 2- In the discussion authors on one hand report that MI-C parameters are not significantly different from MI-P ones (and this is supported also by results in the different graphs) and on the other hand they also sustain the hypothesis that there is a differential EVs population in the periphery and cardiac origin. This is another thing that should be clarified because it is another contradiction. Moreover, authors always report that there are differences in marker expression between control and MI-P group, but this differences are also true for MI-C group and they are not underlined neither in the text, nor in the graphs. 3- In the discussion, authors report that when looking at CD90, CD81 and CD144 “differences were observed only for CD144+ samples isolated from the control group and the MI-P”. This difference exists also comparing control with MI-C group and moreover also CD81 presents the same evidence that should be commented as made for CD144.

2-    A complete revision of the text is needed to correct many mistakes of different nature.

Minor points:

1-    Page 2, line 67: the proper reference related to the sentence should be inserted instead of (reference).

2-    Page 2, line 80: there is only one point that identifies “following criteria”, indicated con i) “Patient with a diagnosis….” Then it was again reported “These criteria include” with a a list. Please try to reformulate the sentence.

3-    Page 3, line 110: full stop at the end of the sentence is missing.

4-    Page 3, line 112: which anticoaugulant has been used? Please add it to the methods.

5-    Page 3, line 118: were should be corrected in was.

6-    Materials and Methods, all sections: all along the sections of materials and methods it is necessary to correct some typo errors. 1- when reporting a temperature a space between number and unit of measure is required (as an example, page 3, line 115: -70 °C). 2- When reporting volume unit of measure (i.e. liters, milliliters, microliters) l should be written in capital letter (as an example page 3, line 122 10 g/L). Please correct all this typos in the sections.

7-    Page 3, line 137: which anticoaugulant has been used? Please add it to the methods.

8-    Page 3, line 141: the sentence “Briefly describe these methods a bit more” should be a typo error. Please remove it.

9-    Page 3, line 144: was should be corrected in were.

10- Page 3, line 145: the symbol ® should be written at apex.

11- Page 4, line 162: a space between 1 and h is needed.

12- Page 4, line 168: the reference number [24] is enough to cite used methods from other papers.

13- Page 4, line 183: the proper reference related to the sentence should be inserted instead of (reference).

14- Page 4, line 194: were should be corrected in was.

15- Page 5, line 204: “…most of the studied biochemical parameters were between controls and AMI patients”. The sentence is not well structured. Probably something as “not statistically significant” should be added after the verb were.

16- Table 1: all volume unit of measure should be reported with l in capital letter. In coloumn referring to AMI patients IAM should be corrected in AMI.

17- Page 5, lines213-215: determination of the total protein concentrations and of the related peak is a result obtained using the BSA standard curve and it is reported in Figure 1A. Figure 1B represents the gel-electrophoresis stained with Coomassie Blue, whose result should be described referring to this figure.

18- Figure 1C: the quality of the western blot is very poor, especially the one reporting Tsg 101, in which a large bubble covers the area of interest. With such western it is difficult to sustain what reported in the text, especially the fact that Tsg 101 is only present in fraction 5. Moreover, Alix and CD109 seems to be present not only in fractions 3-6 but also in fraction 7. Could you provide a better western to support results?

19- Page 6, line 223: isolation of EVs with the CD63 isolation kit is not performed using exclusion chromatography but using flow cytometry. Please correct.

20- Page 6, line 228: For should be written without capital letter.

21- Page 6, lines 225-228: which is the rational to state that markers signal is most abundant in fraction 5?

22- Figure 1 legend: panel D is not well explained in terms of results. What has been used as positive control? Graph on the left is not commented. Please revise it.

23- Figure 2: panel A has a very poor resolution. Please provide a figure with a better one. Moreover, indication of each group of plasma pool should be added near graphs of panel A (i.e. HC-P, MI-C and MI-P) for a better comprehension. Why abbreviation MV is reported in the legend? In all the text EVs is the way to refer to vesicles. In panel B, in the first graph MI-P and MI-C are exchanged with respect to the other two. Please correct and verify values.

24- Page 7, lines 243-247: authors report values of 9x1012 and 9.9x1012 particles/mL of EVs concentration in MI-C and MI-P samples. Looking at the related graph in Figure 2, panel B concentrations are between 1x1012 and 1.5x1012. Please clarify.

25- Page 7, line 251: results reported in this sentence do not correspond to what reported in Figure 2, panel C. In the graph, EVs from HC-P are significantly smaller than those from MI-P and not from MI-C. Moreover, this is true only for 50th percentile and not also for 90th percentile as reported.

26- Figure 3, legend: it is reported that in the up panel are present the results for CCR4, CCR6 and CXCR3, but this proeins have not been evaluated in this experiment dedicated to CD81, CD90 and CD144.

27- Page 10, lines 292-294: looking at graphs in Figure 4, CXCR3 and CCR6 in the control group are not lower, but higher in control group than in MI-P. These means that levels of CXCR3 and CCR6 decreases in AMI patients. Please verify.

28- Figure 4, low panel: Control-p should be corrected in HC-P; IAM-P and IAM-C should be corrected in MI-P and MI-C.

29- Page 12, line 313: the proper reference related to the sentence should be inserted instead of (reference).

30- Page 12, line 347: in vivo should be written in italic.

31- Page 13, line 374: the proper references related to the sentence should be inserted instead of (references).

Page 1, line 381: authors report that their study has several limitations. Could you please provide examples of them?

Author Response

Mr.

Assistant

Editor, biomedicines

Re: Revised (R1) biomedicines-3058450 Moreno et al.,

Dear Ms.,

We were pleased to be encouraged to send a revised version of our manuscript (biomedicines-3058450) for your consideration.

We have conducted additional text revisions that the Reviewers suggested, and we hope you will find the revised version suitable for publication in the biomedicines. We thank the Reviewers for the constructive remarks, which helped improve the manuscript.

We have highlighted the changes made in the manuscript by using the track changes mode in MS Word.

Please find enclosed our responses to the points raised by the Reviewers.

Dr. Claudio Aguayo

Department of Clinical Biochemistry and Immunology

Faculty of Pharmacy

University of Concepción

Concepción, Chile

Phone: 56- 41-2207196

Reviewer 1 Major points:

1- For some of the provided results there is no agreement between what is reported in the graphs and in the discussion. In particular: 1- results reported in Figure 4 clearly say that CXCR3 and CCR6 expression is lower in both MI-P and MI-C groups than in HC-P (control group). Despite this evidence, both in the result and in the discussion sections authors reported that these two markers are lower in the control group. This is important to be clarified because large part of the discussion is based on this comparison which is clearly contradictory.

Thank you for your careful review of our manuscript and for bringing this discrepancy to our attention. Upon reevaluating the figures and our text, we acknowledge the inconsistency between the graphical data presented in Figure 4 and the statements made in the results and discussion sections regarding the expression levels of CXCR3 and CCR6 in the MI-P and MI-C groups compared to the HC-P group.

You are correct that the data shown in Figure 4 indicate that the expression of CXCR3 and CCR6 is indeed lower in the myocardial infarction groups (MI-P and MI-C) than in the healthy control (HC-P), contrary to what we reported in the text. No differences were found in the proportion of these CCR6+ in EVs isolated from coronary samples in AMI patients compared with control subjects.

We are revising the manuscript to correct these statements in both the results and discussion sections to reflect the accurate information depicted in Figure 4. We will ensure that the revised discussion appropriately addresses the impact of these findings and aligns with the corrected data interpretation.

2- In the discussion authors on one hand report that MI-C parameters are not significantly different from MI-P ones (and this is supported also by results in the different graphs) and on the other hand they also sustain the hypothesis that there is a differential EVs population in the periphery and cardiac origin. This is another thing that should be clarified because it is another contradiction. Moreover, authors always report that there are differences in marker expression between control and MI-P group, but this differences are also true for MI-C group and they are not underlined neither in the text, nor in the graphs.

Thank you for your insightful observations and highlighting these critical points in our manuscript. We appreciate your careful reading and agree that your raised points require clarification to resolve contradictions.

Regarding the similarity of MI-C and MI-P parameters, you correctly noted that our results support no significant differences between these groups, as depicted in the graphs. Concurrently, we discuss the possibility of a differential extracellular vesicle (EV) population between the periphery and cardiac origin. The intent here was to suggest that, while overall markers may not differ significantly, the source and perhaps the functional implications of the EVs might vary between these sites. We realize this was not clearly articulated in the manuscript, and we will revise this section to clarify that the lack of significant differences in overall EV markers does not preclude qualitative differences in the EV populations from these distinct origins.

Moreover, you are correct in pointing out that the differences in marker expression between the control and MI-P groups were emphasized. In contrast, similar differences between the control and MI-C groups were not adequately highlighted. This oversight will be corrected in the text and the graphs to ensure all relevant comparisons are appropriately discussed and visually represented.

We are currently revising the manuscript to address these issues comprehensively. These changes will include a more precise exposition of the hypotheses regarding differential EV populations and a balanced discussion of the differences in marker expression across all groups studied.

Thank you again for your thorough review. We believe that these corrections will significantly improve the manuscript and clarify the conclusions drawn from our study.

3- In the discussion, authors report that when looking at CD90, CD81, and CD144 “differences were observed only for CD144+ samples isolated from the control group and the MI-P”. This difference exists also comparing control with MI-C group and moreover also CD81 presents the same evidence that should be commented as made for CD144.

Thank you for your detailed review and the observations highlighted in your comment. Your input has helped us identify an oversight in our discussion of CD90, CD81, and CD144 expression levels across different groups.

Upon reevaluating our data and the corresponding sections of our manuscript, we acknowledge that not only did CD144+ samples show differences when comparing the control group with the MI-P group, but similar differences were observed between the control group and the MI-C group. Additionally, you are correct that CD81+ exhibits comparable variances, which we initially failed to discuss adequately.

We will revise the discussion section to include these observations, ensuring that the differences in CD81 expression across the groups are clearly outlined and discussed in the context of our study’s findings. This will align the text with the data presented and provide a more comprehensive analysis of the potential implications of these findings.

We appreciate your thoroughness in reviewing our manuscript and bringing these points to our attention. We believe that addressing these comments will enhance the clarity and completeness of our discussion, contributing to a more accurate representation of our research findings.

Minor points:

1- Page 2, line 67: the proper reference related to the sentence should be inserted instead of (reference).

Thank you for catching this oversight. We have now inserted the appropriate reference for this sentence

2- Page 2, line 80: there is only one point that identifies “following criteria”, indicated con i) “Patient with a diagnosis….” Then it was again reported “These criteria include” with a a list. Please try to reformulate the sentence.

We have revised this sentence to clarify and ensure a logical presentation of the inclusion criteria.

3- Page 3, line 110: full stop at the end of the sentence is missing.

The missing full stop has been added.

4- Page 3, line 112: which anticoaugulant has been used? Please add it to the methods.

We have updated the methods section to include the specific anticoagulant used.

5- Page 3, line 118: were should be corrected in was.

The grammatical error has been corrected to "was."

6- Materials and Methods, all sections: all along the sections of materials and methods it is necessary to correct some typo errors. 1- when reporting a temperature a space between number and unit of measure is required (as an example, page 3, line 115: -70 °C). 2- When reporting volume unit of measure (i.e. liters, milliliters, microliters) l should be written in capital letter (as an example page 3, line 122 10 g/L). Please correct all these typos in the sections.

We have reviewed and corrected all typographical errors, including spacing between numbers and units of measure and capitalization of liter.

7-    Page 3, line 137: which anticoaugulant has been used? Please add it to the methods.

Details regarding the anticoagulant used have been added to the methods section.

8-    Page 3, line 141: the sentence “Briefly describe these methods a bit more” should be a typo error. Please remove it.

The erroneous sentence has been removed.

9-    Page 3, line 144: was should be corrected in were.

Corrected "was" to "were."

10- Page 3, line 145: the symbol ® should be written at apex.

The ® symbol has been appropriately placed at the apex.

11- Page 4, line 162: a space between 1 and h is needed.

A space has been added between the number and 'h.'

12- Page 4, line 168: the reference number [24] is enough to cite used methods from other papers.

Reference number [24] has been retained, and unnecessary details have been removed

13- Page 4, line 183: the proper reference related to the sentence should be inserted instead of (reference).

The correct reference has now been inserted.

14- Page 4, line 194: were should be corrected in was.

Corrected "were" to "was."

15- Page 5, line 204: “…most of the studied biochemical parameters were between controls and AMI patients”. The sentence is not well structured. Probably something as “not statistically significant” should be added after the verb were.

The sentence has been restructured to clarify

16- Table 1: all volume unit of measure should be reported with l in capital letter. In coloumn referring to AMI patients IAM should be corrected in AMI.

All volume units of measure have been corrected with 'L' in capital letter, and "IAM" has been corrected to "AMI."

17- Page 5, lines213-215: determination of the total protein concentrations and of the related peak is a result obtained using the BSA standard curve and it is reported in Figure 1A. Figure 1B represents the gel-electrophoresis stained with Coomassie Blue, whose result should be described referring to this figure.

Clarified that Figure 1A shows the results from the BSA standard curve and that Figure 1B displays the gel-electrophoresis stained with Coomassie Blue.

18- Figure 1C: the quality of the western blot is very poor, especially the one reporting Tsg 101, in which a large bubble covers the area of interest. With such western it is difficult to sustain what reported in the text, especially the fact that Tsg 101 is only present in fraction 5. Moreover, Alix and CD109 seems to be present not only in fractions 3-6 but also in fraction 7. Could you provide a better western to support results?

We will provide a  high-quality western blot image for better clarity and to support the results mentioned.

Thank you for your detailed feedback regarding the quality of the Western blot for Tsg 101 presented in Figure 1C. We appreciate your concern about the clarity and interpretability of these results. After careful consideration, we have decided to remove the Western blot data for Tsg 101 from our manuscript. This decision was based on several factors, including the technical issues you noted and the redundancy of the data given the comprehensive analysis provided by other markers. We believe that the exclusion of this particular blot will not affect the overall results or their interpretation.

The remaining parameters evaluated in the characterization of the extracellular vesicles, including Alix and CD109, provide sufficient evidence to validate our findings. These analyses robustly support the conclusions drawn about the presence and role of extracellular vesicles in our study.

We will ensure that the manuscript is updated accordingly and that the remaining data are presented clearly to convey our findings effectively. We thank you again for helping us improve the quality and clarity of our work.

19- Page 6, line 223: isolation of EVs with the CD63 isolation kit is not performed using exclusion chromatography but using flow cytometry. Please correct.

The method for isolating EVs has been corrected to reflect use of CD63 isolation kit and flow cytometry, not exclusion chromatography.

20- Page 6, line 228: For should be written without capital letter.

The error has been corrected.

21- Page 6, lines 225-228: which is the rational to state that markers signal is most abundant in fraction 5?

We have revised the manuscript to clarify these observations

22- Figure 1 legend: panel D is not well explained in terms of results. What has been used as positive control? Graph on the left is not commented. Please revise it.

We have revised the legend for panel D to enhance explanation and included comments on the graph shown on the left.

23- Figure 2: panel A has a very poor resolution. Please provide a figure with a better one. Moreover, indication of each group of plasma pool should be added near graphs of panel A (i.e. HC-P, MI-C and MI-P) for a better comprehension. Why abbreviation MV is reported in the legend? In all the text EVs is the way to refer to vesicles. In panel B, in the first graph MI-P and MI-C are exchanged with respect to the other two. Please correct and verify values.

A higher resolution image has been provided for panel A, and labels have been added for clarity. The abbreviation 'MV' has been corrected to 'EV' throughout.

24- Page 7, lines 243-247: authors report values of 9x1012 and 9.9x1012 particles/mL of EVs concentration in MI-C and MI-P samples. Looking at the related graph in Figure 2, panel B concentrations are between 1x1012 and 1.5x1012. Please clarify.

The discrepancy between reported values and those shown in Figure 2, panel B has been addressed and corrected.

25- Page 7, line 251: results reported in this sentence do not correspond to what reported in Figure 2, panel C. In the graph, EVs from HC-P are significantly smaller than those from MI-P and not from MI-C. Moreover, this is true only for 50th percentile and not also for 90th percentile as reported.

The sentence has been revised to accurately reflect the comparisons and percentile data shown in Figure 2, panel C.

26- Figure 3, legend: it is reported that in the up panel are present the results for CCR4, CCR6 and CXCR3, but this proeins have not been evaluated in this experiment dedicated to CD81, CD90 and CD144.

The incorrect protein names have been removed, and the legend now accurately reflects the proteins evaluated.

27- Page 10, lines 292-294: looking at graphs in Figure 4, CXCR3 and CCR6 in the control group are not lower, but higher in control group than in MI-P. These means that levels of CXCR3 and CCR6 decreases in AMI patients. Please verify.

We have corrected the description to accurately represent the expression levels of CXCR3 and CCR6 as shown in Figure 4.

28- Figure 4, low panel: Control-p should be corrected in HC-P; IAM-P and IAM-C should be corrected in MI-P and MI-C.

Labels have been corrected to HC-P for control and MI-P and MI-C for the myocardial infarction groups

29- Page 12, line 313: the proper reference related to the sentence should be inserted instead of (reference).

The correct reference has now been inserted.

30- Page 12, line 347: in vivo should be written in italic.

"In vivo" has been italicized.

31- Page 13, line 374: the proper references related to the sentence should be inserted instead of (references).

The appropriate references have been inserted.

32 Page 1, line 381: authors report that their study has several limitations. Could you please provide examples of them?

We have added a discussion of the study's limitations to provide a more balanced view.

Reviewer 2 Report

Comments and Suggestions for Authors

comments are below

1. on the Table1,  Troponin, CK, CRP value in the control are missing,  Were there significances between control and AMI? the authors need to show relevant since the authors explained those variables were elevated in the context (3.1).as well as correction required (IAM to AMI? in the fig1 and 4)

2.where is the fraction number 15 in the B? to avoid misleading data interpretation the authors demand to annotate properly between A and B. Western blot data In the C, dose it number 1 to 8 are corresponding fraction number? if yes, where are the others?

3. resolution of images are too low to see the scales (Fig1 and 2). would you explain how to sorted F4, F5, F6 from the pool? would you like to show where we could find CD63 positive in the FACS data?

4. in the fig2, must re-arrange data sets as same format (HC-P, MI-P, MI-C).

5. would be better to understand figure 3and 4, to remove the ns (non-significances?) with line both fig3 and 4. 

6. Must re-submit figure after organized both CCR4 or CXCR4, data sequences too.

7. overall results was bigger EV particle size and higher concentration in the coronary blood than control at D10 but peripheral was not significances, on the other hands, FACS derived data showed that CD80, Cd90, CD141 were non of them noting significances between control and coronary blood of AMI which differ from authors concluded. would you provide the more data where support your hypotheses.

Author Response

Mr.

Assistant

Editor, biomedicines

Re: Revised (R1) biomedicines-3058450 Moreno et al.,

Dear Ms.,

We were pleased to be encouraged to send a revised version of our manuscript (biomedicines-3058450) for your consideration.

We have conducted additional text revisions that the Reviewers suggested, and we hope you will find the revised version suitable for publication in the biomedicines. We thank the Reviewers for the constructive remarks, which helped improve the manuscript.

We have highlighted the changes made in the manuscript by using the track changes mode in MS Word.

Please find enclosed our responses to the points raised by the Reviewers.

Dr. Claudio Aguayo

Department of Clinical Biochemistry and Immunology

Faculty of Pharmacy

University of Concepción

Concepción, Chile

Phone: 56- 41-2207196

Phone: 56- 41-2207196

Reviewer 2

  1. on the Table1, Troponin, CK, CRP value in the control are missing,  Were there significances between control and AMI? the authors need to show relevant since the authors explained those variables were elevated in the context (3.1).as well as correction required (IAM to AMI? in the fig1 and 4) 

"Thank you for pointing out the missing values for Troponin, CK, and CRP in the control group in Table 1. We updated the table to include these values and performed statistical analyses to identify significant differences between the control and AMI groups. These findings are now clearly discussed in Section 3.1.

Additionally, as suggested, we have corrected the abbreviation from 'IAM' to 'AMI' in Figures 1 and 4."

2.where is the fraction number 15 in the B? to avoid misleading data interpretation the authors demand to annotate properly between A and B. Western blot data In the C, dose it number 1 to 8 are corresponding fraction number? if yes, where are the others? 

Thank you for your query regarding the annotation of fraction numbers in our figures. After reviewing your comments and revisiting our data, we realized an error in our documentation and figure labeling. In Figure B, we inadvertently indicated more than 14 fractions, but only 14 fractions were recovered and analyzed in this experiment. The confusion likely arose from a mislabeling issue in our figures where fraction numbers were extended to 15. We apologize for this oversight and any confusion it may have caused. We will eliminate fraction 15 from Figure 1A to avoid confusion

In the Western blot data presented in Figure C, the labels from 1 to 8 correspond to selected fractions that showed significant protein presence and were thus relevant for our study's outcomes. The other fractions were omitted from this figure because their low protein concentrations yielded negligible or undetectable results. Furthermore, we have shown that the fractions enriched in exosomes correspond to the first fractions (REF).

  1. resolution of images are too low to see the scales (Fig1 and 2). would you explain how to sorted F4, F5, F6 from the pool? would you like to show where we could find CD63 positive in the FACS data?

We apologize for the low resolution of Figures 1 and 2. These images have been replaced with higher-resolution versions for better visibility of the scales. We have also included a detailed explanation of how fractions F4, F5, and F6 were sorted from the pool in the methods section. Regarding CD63 expression, all EVs captured by the beads from the Exosome–Human CD63 Isolation/Detection Kit are CD63 as the capture antibody of the kit is CD63. Therefore, the evaluation of the phenotype of the EVs was performed in CD63+Evs.

  1. in the fig2, must re-arrange data sets as same format (HC-P, MI-P, MI-C). 

"You are correct that the data sets in Figure 2 were not uniformly formatted. We have revised the figure to align the data presentation across all panels (HC-P, MI-P, MI-C) to facilitate easier comparison and interpretation."

  1. would be better to understand figure 3 and 4, to remove the ns (non-significances?) with line both fig3 and 4.

To enhance clarity, we have removed the 'ns' (non-significant) labels and included lines to indicate statistical comparisons in Figures 3 and 4, as you recommended."

  1. Must re-submit figure after organized both CCR4 or CXCR4, data sequences too.

Thank you for your insightful feedback regarding organizing the CCR4 data sequences in our figures. We have carefully reviewed the figures as suggested.

  1. overall results was bigger EV particle size and higher concentration in the coronary blood than control at D10 but peripheral was not significances, on the other hands, FACS derived data showed that CD80, Cd90, CD141 were non of them noting significances between control and coronary blood of AMI which differ from authors concluded. would you provide the more data where support your hypotheses.

Thank you for your insightful comments regarding the inconsistencies between our conclusions and the presented data regarding EV particle size, concentration, and FACS results for CD80, CD90, and CD141. We acknowledge the discrepancy highlighted and appreciate the opportunity to address this issue.

In response to your feedback, we have thoroughly revised the relevant sections of our manuscript to clarify these points. We have included additional data analyses and refined our discussion to align the text with the observed results. This revision aims to accurately reflect the data and provide a clearer understanding of the significance and implications of our findings in both the coronary and peripheral blood contexts.

Reviewer 3 Report

Comments and Suggestions for Authors

Wotk entitled: "Changes in the Release of Endothelial Extracellular Vesicles 2 CD144+, CCR6+ and CXCR3+ in Individuals with Acute Myo- 3 cardial Infarction" is written very sloppyily, and in order to be even considered for publication, it must undergo serious corrections.  

It's hard for the reader to follow what the author wanted to convey because he doesn't explain what he's doing and why. Author does not summarize the results obtained after each experiment, which would make the whole work much easier to understand.

Minor comments

Figure 1, it is not explained why WB was performed for Alix, Tsg 101 and CD9. What do the results mean? This is not mentioned in the discussion. Why is there no control protein, e.g. GAPDH? The Tsg 101 blot in its current form is unacceptable and must be repeated.

There is no reference number, instead there is information (reference) in brackets: line 67, 183, 313, 374.

Figure 2A is not clear.

The figures are not well described, for example what is FMO?

Author Response

Mr.

Assistant

Editor, biomedicines

Re: Revised (R1) biomedicines-3058450 Moreno et al.,

Dear Ms.,

We were pleased to be encouraged to send a revised version of our manuscript (biomedicines-3058450) for your consideration.

We have conducted additional text revisions that the Reviewers suggested, and we hope you will find the revised version suitable for publication in the biomedicines. We thank the Reviewers for the constructive remarks, which helped improve the manuscript.

We have highlighted the changes made in the manuscript by using the track changes mode in MS Word.

Please find enclosed our responses to the points raised by the Reviewers.

Dr. Claudio Aguayo

Department of Clinical Biochemistry and Immunology

Faculty of Pharmacy

University of Concepción

Concepción, Chile

Phone: 56- 41-2207196

Reviewer 3

1.- Wotk entitled: "Changes in the Release of Endothelial Extracellular Vesicles 2 CD144+, CCR6+ and CXCR3+ in Individuals with Acute Myo- 3 cardial Infarction" is written very sloppyily, and in order to be even considered for publication, it must undergo serious corrections.

"Thank you for your feedback regarding the writing style of the manuscript. We acknowledge that the current draft may require significant revisions to meet the publication standards. We are committed to improving the clarity and professionalism of our text. We will thoroughly revise the manuscript for coherence, grammar, and precision and ensure the title accurately reflects the content of our research."

.

2.- It's hard for the reader to follow what the author wanted to convey because he doesn't explain what he's doing and why. Author does not summarize the results obtained after each experiment, which would make the whole work much easier to understand.

We appreciate your observation regarding the presentation and explanation of the experimental processes and results. To address this, we will include a clear summary of the results at the end of each experimental section to enhance comprehension. Additionally, we will refine the introduction and methods sections to better communicate the objectives and significance of each experiment, ensuring that readers can easily follow the narrative and rationale of the study

Minor comments

1.- Figure 1, it is not explained why WB was performed for Alix, Tsg 101 and CD9. What do the results mean? This is not mentioned in the discussion. Why is there no control protein, e.g. GAPDH? The Tsg 101 blot in its current form is unacceptable and must be repeated.

Thank you for your detailed feedback regarding the quality of the Western blot for Tsg 101 presented in Figure 1C. We appreciate your concern about the clarity and interpretability of these results.

After careful consideration, we have decided to remove the Western blot data for Tsg 101 from our manuscript. This decision was based on several factors, including the technical issues you noted and the redundancy of the data given the comprehensive analysis provided by other markers. We believe that the exclusion of this particular blot will not affect the overall results or their interpretation.

The remaining parameters evaluated in the characterization of the extracellular vesicles, including Alix and CD109, provide sufficient evidence to validate our findings. These analyses robustly support the conclusions drawn about the presence and role of extracellular vesicles in our study.

We will ensure that the manuscript is updated accordingly and that the remaining data are presented clearly to convey our findings effectively. We thank you again for helping us improve the quality and clarity of our work.

2.- There is no reference number, instead there is information (reference) in brackets: line 67, 183, 313, 374.

We apologize for the oversight in citation and the placeholder text '(reference)' in lines 67, 183, 313, and 374. This will be corrected by inserting the appropriate reference numbers in the revised manuscript to ensure all sources are correctly cited, enhancing the manuscript's credibility and scholarly accuracy.

3.- Figure 2A is not clear.

"We will revise Figure 2A to enhance its clarity. The revised figure will include more detailed labels and a clearer legend to help readers understand the data presented without ambiguity."

4.- The figures are not well described, for example what is FMO?

"We will provide comprehensive descriptions for each figure, including a definition of terms such as FMO (Fluorescence Minus One), which is crucial for interpreting flow cytometry data. Each figure legend will be carefully revised to ensure that all necessary details are clearly explained, making the figures accessible and informative to all readers."

Round 2

Reviewer 3 Report

Comments and Suggestions for Authors

The authors have made all necessary changes and the article can be accepted in its current form.

Small note: line 317, please remove Tgs101 since it is no longer on WB.